# A Gas Path Fault Contribution Matrix for Marine Gas Turbine Diagnosis Based on a Multiple Model Fault Detection and Isolation Approach

**Qingcai Yang [1], Shuying Li [1], Yunpeng Cao [1,*] , Fengshou Gu [2,*] and Ann Smith [2]**

[1] College of Power and Energy Engineering, Harbin Engineering University, Harbin 150001, China; yangqingcai@hrbeu.edu.cn (Q.Y.); lishuying@hrbeu.edu.cn (S.L.)

[2] Centre for Efficiency and Performance Engineering, University of Huddersfield, Huddersfield HD1 3DH, UK; a.smith@hud.ac.uk

[*] Correspondence: caoyunpeng@hrbeu.edu.cn (Y.C.); f.gu@hud.ac.uk (F.G.); Tel.: +44-148-447-3548 (F.G.)

**Abstract:** To ensure reliable and efficient operation of gas turbines, multiple model (MM) approaches have been extensively studied for online fault detection and isolation (FDI). However, current MM-FDI approaches are difficult to directly apply to gas path FDI, which is one of the common faults in gas turbines and is understood to mainly be due to the high complexity and computation in updating hypothetical gas path faults for online applications. In this paper, a fault contribution matrix (FCM) based MM-FDI approach is proposed to implement gas path FDI over a wide operating range. As the FCM is realized via an additive term of the healthy model set, the hypothetical models for various gas path faults can be easily established and updated online. In addition, a gap metric analysis method for operating points selection is also proposed, which yields the healthy model set from the equal intervals linearized models to approximate the nonlinearity of the gas turbine over a wide range of operating conditions with specified accuracy and computational efficiency. Simulation case studies conducted on a two-shaft marine gas turbine demonstrated the proposed approach is capable of adaptively updating hypothetical model sets to accurately differentiate both single and multiple faults of various gas path faults.

**Keywords:** gas turbine; gas path fault; fault detection and isolation; multiple model; gap metric analysis

## 1. Introduction

With increasing gas turbine complexity, requirements for ensuring reliability and security are growing significantly, and diagnosis of gas path component faults is becoming a major issue [1]. The main causes of gas path faults include fouling, erosion, corrosion, and foreign object damage (FOD), which degrade the performance of the gas turbine, further reduce its safety and stability, and lead to decreased fuel economy and increases in operation and maintenance costs [2]. Gas path fault diagnosis approaches aim to detect and locate a fault in time to prevent a catastrophic consequence, and therefore contribute to the development of a maintenance schedule. A large number of gas path fault diagnosis approaches have been proposed and achieved good results in practical applications, such as neural networks [3–6], gas path analysis derivatives [7], genetic algorithms [8], and adaptive estimation [9–11]. Fault detection and isolation (FDI), as the core of the gas turbine fault diagnosis, has received extensive attention from researchers [12–15].

The multiple model (MM) approach is highly regarded due to its ability to simultaneously ensure high levels of diagnosis accuracy whilst maintain acceptable computational costs. It is based on

hypothesis testing and conditional probability, which transforms a complex problem into several simple problems to detect and isolate the fault [16]. The MM approach was pioneered by Magill [17]. Up to now, three generations of the MM approach have been designed, including autonomous MM, interacting MM, and variable structure MM. A detailed description of these approaches can be found in References [16,18]. Maybeck applied the MM approach to a gas turbine to detect and isolate sensor and actuator faults [19–22], and Khorasani applied the MM approach to gas path fault detection and isolation for jet engines [23–25]. In Reference [23], a nonlinear MM-FDI approach based on the extended Kalman filter (EKF) and the unscented Kalman filter (UKF) was proposed for a jet engine. Then, the MM-FDI approach was used to detect and isolate the gas path fault, and a hierarchical architecture was developed that enabled the detection and isolation of both single and multiple faults in the jet engine [24].

Gas path fault, actuator fault, and sensor fault are three types fault of a gas turbine. Application of the MM-FDI approach in gas path fault identification is still under development. One of the main problems is the design of hypothetical fault models [26], which have a significant impact on the performance of the MM-FDI approach. The gas turbine healthy model set can be described by a set of linear models in Equation (1):

$$\begin{aligned} x_{k+1} &= A_i x_k + B_i u_k + w_k \\ y_k &= C_i x_k + v_k \end{aligned}$$ (1)

where $i$ is an operating point, $i = 1, 2, \dots, M$; $x \in n \times 1$, $y \in l \times 1$, and $u \in t \times 1$ denote the state vector, measurement vector, and control vector, respectively; and $n$, $l$, and $t$ denote the dimension of the state vector, measurement vector, and control vector, respectively. $w_k \in n \times 1 \sim (0, Q)$ and $v_k \in l \times 1 \sim (0, R)$ are independent discrete-time process noises and measurement noises, respectively; and $A_i \in n \times n$, $B_i \in n \times t$, and $C_i \in l \times n$ denote the state matrix, control matrix, and output matrix of the healthy condition at the operating point $i$, respectively.

Faults in the sensors cause changes in the measurement vector $y$, and faults in the actuators cause changes in the control vector $u$. Therefore, hypothetical models for sensor and actuator faults can be established by directly changing the corresponding measurement vector and control vector of the healthy model in Equation (1) [18,22,24]. However, the consequences of gas path faults due to component defects cause changes in the system matrices $A$, $B$, and $C$, hence the traditional gas path fault model can be described in the form of Equation (2):

$$\begin{aligned} x_{k+1} &= A_i' x_k + B_i' u_k + w_k \\ y_k &= C_i' x_k + v_k \end{aligned}$$ (2)

where $A_i' \in n \times n$, $B_i' \in n \times t$, and $C_i' \in l \times n$ are the system matrices of a gas path fault condition at the operating point $i$.

According to Equation (2), each gas path fault at the operating point $i$ is matched to a hypothetical fault model, which means that the system matrices corresponding to each gas path fault need to be established separately in advance for applying the MM-FDI approach. Therefore, establishing the hypothetical fault models is a very time-consuming task, and they are also not easy to update automatically online. Although the proposed hierarchical architecture can reduce the number of models of each level [24], due to the stochasticity and diversity of fault occurrences, the establishment of the most possible hypothetical fault models will face combinatorial explosion problems, which make the MM-FDI approach difficult to implement practically.

In this paper, a gas path fault contribution matrix (FCM) was introduced to improve the performance of the MM-FDI approach to address the above problem, and a systemic method to select the operating points that form the healthy model set was carried out based on gap metric analysis to achieve MM-FDI over a wide range of operating conditions. To evaluate the effectiveness of the FCM based MM-FDI approach, several simulation case studies on a two-shaft marine gas turbine

were conducted. Performance of the proposed approach for FDI and the influence of the number of available measurements and measurement outliers was analyzed for single and multiple fault cases.

The remainder of this paper is organized as follows: Details pertaining to the FCM based MM-FDI approach are introduced in Section 2. In Section 3, a gap metric analysis method is introduced to yield the healthy model set. Section 4 shows the application of the proposed approach to a two-shaft marine gas turbine, and the results of the simulation case studies are analyzed and discussed. Finally, the main conclusions are presented in Section 5.

## 2. The FCM Based MM-FDI Approach

### 2.1. Model Set Design Based on FCM

Inspired by the estimation algorithm used in adaptive performance optimization of turbofan engines [27], this paper introduces the gas path fault contribution matrices $E$ and $D$ into Equation (1) as additive terms of the healthy model, so that they can be adjusted easily to balance the deviations caused by gas path faults. This gas turbine model set design based FCM can be represented in Equation (3):

$$
\begin{cases}
x_{k+1} = Ax_k + \begin{bmatrix} B & E \end{bmatrix} \begin{bmatrix} u_k \\ f_k \end{bmatrix} \\
y_k = Cx_k + \begin{bmatrix} 0 & D \end{bmatrix} \begin{bmatrix} u_k \\ f_k \end{bmatrix}
\end{cases}
\tag{3}
$$

where the matrices $E \in n \times q$ and $D \in l \times q$ denote FCMs which represent the effect of gas a path fault on the healthy model, $q$ represents the number of gas path faults in the gas turbine, $f_k = \sum_{i=1}^{q} b_i z_i$ is the gas path fault vector, $b_i$ is the fault amplitude, and $z_i$ is the location of $i-$th fault. In particular, $f_k$ is a $q \times 1$ zero-valued vector for the healthy condition, and it becomes a nonzero value when a gas path fault has occurred.

In Equation (3), the $j$-th hypothetical model at the operating point $i$ can be represented by

$$
\begin{cases}
x_{k+1}^j = A_i x_k^j + \begin{bmatrix} B_i & E_i \end{bmatrix} \begin{bmatrix} u_k^j \\ f_k^j \end{bmatrix} + w_k \\
y_k^j = C_i x_k^j + \begin{bmatrix} 0 & D_i \end{bmatrix} \begin{bmatrix} u_k^j \\ f_k^j \end{bmatrix} + v_k
\end{cases}
\tag{4}
$$

As shown in Equation (4), different gas path hypothetical fault models can be established by directly quantifying the value of the corresponding gas path fault matrix $f_k$, which is similar to the effect of the control vector $u_k$ on the actuator fault model.

### 2.2. Model Conditional Filtering

The Kalman filter algorithm for the $j-$th hypothetical model at the $i-$th operating point is shown in Equation (5).

$$
\begin{aligned}
&Time-update: \\
&\hat{x}_{k+1|k}^{(j)} = A_i \hat{x}_{k|k}^{(j)} + \begin{bmatrix} B_i & E_i \end{bmatrix} \begin{bmatrix} u_k^{(j)} & f_k^{(j)} \end{bmatrix}^{\mathrm{T}} \\
&\hat{P}_{k+1|k}^{(j)} = A_i P_{k|k}^{(j)} A_i^{\mathrm{T}} + Q \\
&Measurment-update: \\
&K_k^{(j)} = P_{k+1|k}^{(j)} C_i^{\mathrm{T}} [C_i P_{k+1|k}^{(j)} C_i^{\mathrm{T}} + R]^{-1} \\
&\hat{x}_{k+1|k+1}^{(j)} = \hat{x}_{k+1|k}^{(j)} + K_k^{(j)} [y_k - (C_i \hat{x}_{k+1|k}^{(j)} + \begin{bmatrix} 0 & D_i \end{bmatrix} \begin{bmatrix} u_k^{(j)} & f_k^{(j)} \end{bmatrix}^{\mathrm{T}})] \\
&P_{k+1|k+1}^{(j)} = \hat{P}_{k+1|k}^{(j)} - K_k^{(j)} C_i \hat{P}_{k+1|k}^{(j)}
\end{aligned}
\tag{5}
$$

where the Kalman filter algorithm contains two steps, time update and measurement update; therein, $\hat{x}_{k|k}$ and $\hat{x}_{k+1|k+1}$ are the system state estimate at time $k$ and $k+1$, respectively; $\hat{x}_{k+1|k}$ is the system state predict at time $k+1$; $P_{k|k}$ and $P_{k+1|k+1}$ are the state estimate error covariance at time $k$ and $k+1$, respectively; $\hat{P}_{k+1|k}$ is the state predict error covariance at time $k+1$; and $K_k$ is the Kalman gain at time $k$.

### 2.3. Model Probability Update

The conditional probability $\mu_i(k)$ is used to represent the approximation of each hypothetical model to the gas turbine actual condition. The gas turbine actual conditions $AC$ match $Hm_i$ when a given measurement vector $y_k$ is at a discrete time $k$; that is

$$\mu_i(k) = \Pr[AC = Hm_i | y(t_k) = y_k] \tag{6}$$

The conditional probability at a given time $k$ of the hypothetical model set can be recursively calculated by the Bayesian Law according to the conditional probability value of the previous moment and the Gaussian probability density corresponding to the current filter residual, as shown in Equation (7).

$$\mu_i(k) = \frac{f(y_k | Hm_i, y_{k-1}) \mu_i(k-1)}{\sum\limits_{j=1}^{W} f(y_k | Hm_j, y_{k-1}) \mu_j(k-1)} \tag{7}$$

where $f(y_k | Hm_i, y_{k-1})$ denotes the conditional probability density of the $i-$th hypothetical model when the measurements are $y_k$, and $\mu_i(k-1)$ denotes the conditional probability of the $i$-th hypothetical model at the time $k-1$. The Gaussian conditional probability density function is shown in Equation (8).

$$f(y_k | Hm_j, y_{k-1}) = \frac{1}{(2\pi)^{l/2} \left| S_k^j \right|^{1/2}} \exp[-\frac{1}{2}(\gamma_k^j)^T S_k^j \gamma_k^j] \tag{8}$$

where $l$ is the dimension of the measurement parameters, and $S_k^j$ and $\gamma_k^j$ are the innovation covariance and the innovation of the filter corresponding to the $j-$th hypothetical model, respectively, which can be obtained in the filtering process.

### 2.4. Fault Detection and Isolation

The conditional probability of the hypothetical model set can be used as an indication of a fault, as it provides a meaningful measure of how likely each fault is at a given time [28]. Therefore, the actual fault can be detected and isolated by using the maximum probability criteria, as shown in Equation (9):

$$j = \arg \max_{i=1...W} \mu_i(k) > \mu_{TH} \tag{9}$$

where $j$ represents that the $j-$th model is closest to the actual condition, and $\mu_{TH}$ is a preset detect and isolate threshold. To ensure a fast response to changes due to fault effects, a minimum $\mu_i = 0.001$ was used for each model [27]. Additionally, the preset detection and isolation threshold $\mu_T$ was set to 0.98 to reduce the amount of warnings [22].

### 2.5. Hypothetical Model Update

In practice, multiple faults exist in which a second fault occurs after the first fault. In this scenario, no hypothetical model in the initial hypothetical model set matches the actual fault condition. In addition, a gas path abrupt fault with large amplitude may also result in incorrect detection. Therefore, to maintain the performance of the proposed approach, it is a key issue to update the hypothetical model set to detect multiple faults after detecting the first fault, or to detect faults with larger amplitude.

Assuming the initial hypothetical model set is constant and has $W$ hypothetical models, the demonstration of the hypothetical model set update process is shown in Figure 1; the initial hypothetical model set is shown in the left of the figure, where the subscript $j-1$ denotes the $(j-1)-$th element of $f_k$. The initial fault amplitude is assumed to have a deviation of $-1\%$ ($-3\%$ is considered to be severe in practice [29,30]) for each hypothetical fault model. If the hypothetical model $j$ has the largest conditional probability, and it is in excess of the preset threshold $\mu_{TH}$, then the first fault is detected and isolated as the hypothetical model $j$. Subsequently, the initial hypothetical model set will be updated to detect whether or not the current condition is multiple faults or a single fault that is more serious. In the hypothetical model set update process, each hypothetical model should contain the first detected fault; the $j-$th hypothetical model represents a more serious fault ($-2\%$), as shown in the right of Figure 1. In the case of more faults, the update process of the hypothetical model set is in the same manner.

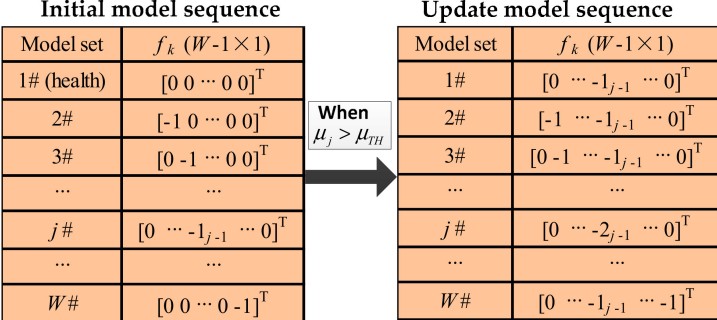

**Figure 1.** Demonstration of the hypothetical model set update.

The detailed hypothetical model set update algorithm is shown in Table 1. First, the detected hypothetical model is determined to obtain the current fault type and the number of faults, and determine the matrix $F_{updata}$ corresponding to the updated hypothetical model set. Then, each column of $F_{updata}$ is substituted into Equation (4) to obtain the updated hypothetical model set. After the hypothetical model set is updated, the conditional probability of each model and the filter will be reinitialized.

**Table 1.** The hypothetical mode set update algorithm.

| |
|---|
| **Step 1: Determine the type and number of the detected faults** |
| $m_i = find(\mu > 0.98)$<br>$z_i = find(F(:, m_i) < 0)$<br>$n = numel(z_i)$ |
| **Step 2: Determine the matrix $F_{update}$ corresponding to the updated hypothetical model set** |
| $F_{update} = F;$<br>$for\ k = 1:n$<br>$\quad F_{update}(z_i(k), :) = F(z_i(k), m_i);$<br>$\quad F_{update}(z_i(k), z_i(k) + 1) = F(z_i(k), m_i) - s_i;$<br>$end$ |
| **Step 3: Substituting each column of $F_{update}$ into Equation (4) to obtain the updated mode set** |
| $\begin{cases} x_{k+1} = Ax_k + \begin{bmatrix} B & E \end{bmatrix} \begin{bmatrix} u_k \\ f_k \end{bmatrix} \\ y_k = Cx_k + \begin{bmatrix} 0 & D \end{bmatrix} \begin{bmatrix} u_k \\ f_k \end{bmatrix} \end{cases}$ |
| **Notation** |
| $\mu$ denotes the conditional probability of the $W$ hypothetical models, a $1 \times W$ vector;<br>$F$ denotes a matrix which is composed of $f_k$ corresponding to each hypothetical model in the current hypothetical model set, $F = \begin{bmatrix} f_k^{1\#} & f_k^{2\#} & \cdots & f_k^{W\#} \end{bmatrix}$, a $q \times W$ matrix;<br>$F_{updata}$ denotes a matrix which is composed of $f_k$ corresponding to each hypothetical model in the updated model set, a $q \times W$ matrix;<br>$s$ denotes the fault amplitude changes during model set updating, s = $-1$. |

*2.6. Implementation of FCM Based MM-FDI*

The overall flow diagram of FCM based MM-FDI is shown in Figure 2, which consists of following steps: Establishing a gas turbine model set based on FCM that can represent system structure at the healthy condition, and where all parameters use the percentage change in the model set. Then, based on a hypothetical model set update algorithm, a bank of Kalman filters corresponding to the hypothetical model set which represents hypothetical conditions of the gas turbine is designed. The conditional probability of each hypothetical model is recursively calculated according to the filter residuals and Bayesian law. Finally, the model that most closely aligns with the actual condition of the gas turbine can be detected and isolated using the maximum probability criteria. Compared with the MM-FDI algorithm in References [23,24], the FCM based MM-FDI approach proposed in this paper solves the problem of gas path fault model set establishment and automatic update of the hypothetical model set.

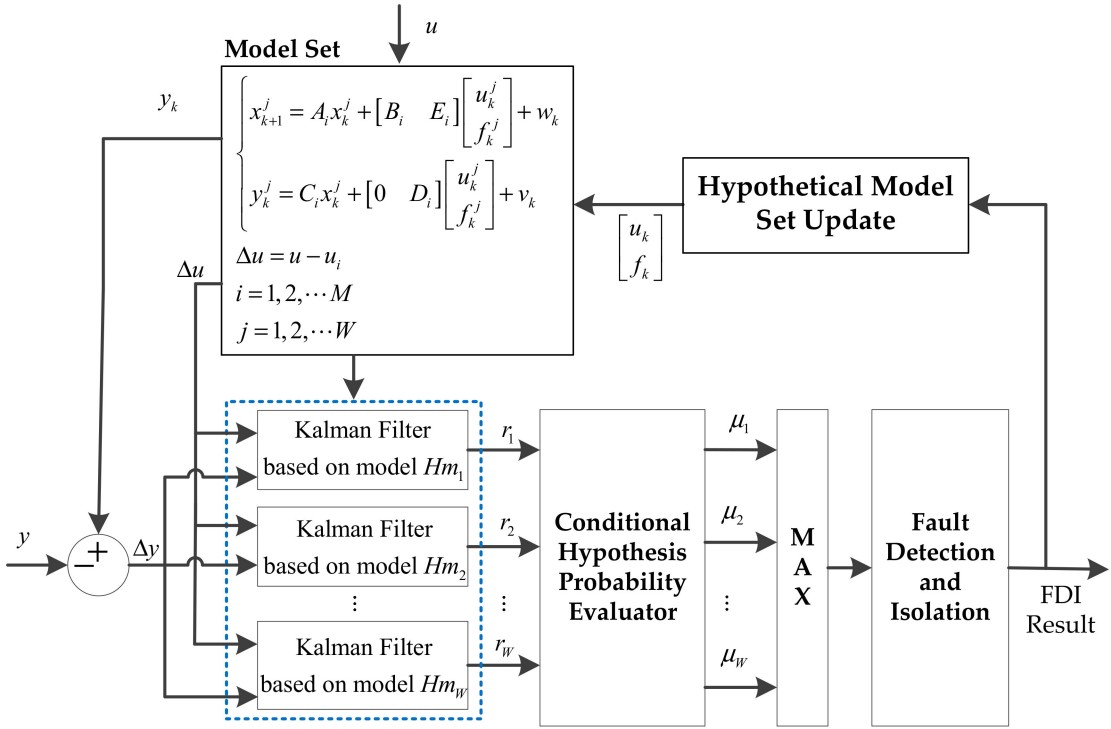

**Figure 2.** Diagram of the fault contribution matrix (FCM) based multiple model fault detection and isolation (MM-FDI) approach.

## 3. Selecting Operating Points Based on Gap Metric Analysis

Selecting the few operating points to form the healthy model set over a wide operating condition is another issue of the MM approach. It usually uses equal intervals or control command [24], which may lead to the redundant or incomplete selection of the linear model. To enhance the computational efficiency, the trajectory piecewise-linear method [31] and range reduction techniques [32] have been proposed for the optimal selecting of operating points.

In this paper, a systemic approach for selecting the operating points of the gas turbine linearized models was carried out based on gap metric analysis. EI-Sakkary [33] pioneered a gap metric that can be used as an indication of the approximation of two linear systems. For two linear models $L_1$ and $L_2$ with the same input and output, the gap between two linear models is [34]

$$\delta(L_1, L_2) = \max\left( \vec{\delta}(L_1, L_2), \vec{\delta}(L_2, L_1) \right) \qquad (10)$$

where $\vec{\delta}\left(L_i, L_j\right)$ is

$$\vec{\delta}\left(L_i, L_j\right) = \inf_{Q \in H_\infty} \left\| \begin{pmatrix} M_i \\ N_i \end{pmatrix} - \begin{pmatrix} M_j \\ N_j \end{pmatrix} Q \right\|_\infty \tag{11}$$

where $i, j = \{1, 2\}, i \neq j$. $Q \in H_\infty$, and $M_i$ and $N_i$, are normalized right coprime factorizations of $L_i$, and defined as

$$L_i = N_i M_i^{-1}, i = 1, 2 \tag{12}$$

According to the gap metric, the range of the gap metric for two linear models is [0, 1], and the smaller the gap metric, the closer the dynamic response between the two linear models. Therefore, when the gap metric of the two linear models is less than the preset threshold $\delta_{TH}$, it means that just one of the models is sufficient, as the other one has become redundant. Otherwise, the two models are not redundant and need to be selected simultaneously.

In this paper, a set of operating points with equal intervals and number of $z$ are selected in the range of the control variable $u$, and the interval $\varphi$ between any two adjacent operating points $u_p$ and $u_{p+1}$ ($p = 1, 2, \ldots, z - 1$) satisfies Equation (13):

$$\varphi = \left| u_p - u_{p+1} \right| \leq \delta u \tag{13}$$

where $\delta u$ is a sufficiently small value such that the gap metric of the linearization models corresponding to any two adjacent points is less than the preset threshold $\delta_{TH}$.

Correspondingly, a set of linearized models $L_i$ ($i = 1, \ldots, z$) corresponding to $z$ operating points was established. Moreover, the gap metric between any two linearized models can be calculated. Finally, the gap metric matrix $G$ of all linearized models is obtained in Equation (14):

$$G = \begin{bmatrix} \delta(L_1, L_z) & \delta(L_2, L_z) & \cdots & \delta(L_z, L_z) \\ \vdots & \vdots & & \vdots \\ \delta(L_1, L_2) & \delta(L_2, L_2) & \cdots & \delta(L_z, L_2) \\ \delta(L_1, L_1) & \delta(L_2, L_1) & \cdots & \delta(L_z, L_1) \end{bmatrix} \tag{14}$$

where for the matrix $G \in z \times z$, $\delta(L_i, L_j) = \delta(L_j, L_i)$, $i, j = 1, 2, \ldots, z$, and $\delta(L_i, L_j) = 0$, when $i = j$. $i = 1$ defines the first operating point.

According to the preset threshold $\delta_T$ and the obtained gap metric matrix $G$, a set of operating points, from which corresponding linearized models form the healthy model set, can be selected from the $z$ operating points. It can be seen from Equation (14) that the $i-$th column in matrix $G$ is the gap metric between the $i-$th linearized model and $z$ linearized models. We keep $i$ unchanged and $j$ increases sequentially, then compare $\delta(L_i, L_j)$ with the preset threshold $\delta_{TH}$. If $\delta(L_i, L_j) \leq \delta_{TH}$, then the linearized model at an operating point $j$ can be replaced by the linearized model at the operating point $i$. Repeating the above process until $\delta(L_i, L_j) > \delta_{TH}$ or $L_j = L_z$, then the $j - 1 - i$ linearized models from the operating point $i$ to $j - 1$ can be represented by the linearized model at the operating point $i$. Meanwhile, the linearized model at the operating point $j$ is the next selected operating point.

## 4. Simulation Results and Discussion

In this section, the proposed FCM based MM-FDI approach is applied to a two-shaft marine gas turbine. Several case studies were conducted to evaluate the performance of the proposed approach, which include single fault detection and isolation, multiple fault detection and isolation, and the influence of the number of available measurements and the measurements' outliers.

The general layout of the two-shaft marine gas turbine is shown in Figure 3 [35]. The gas path components include a compressor, a combustion chamber, a compressor turbine, and a power turbine, wherein the power turbine drives the propeller through the reduction gearbox, which is simplified as a load, as shown in Figure 3.

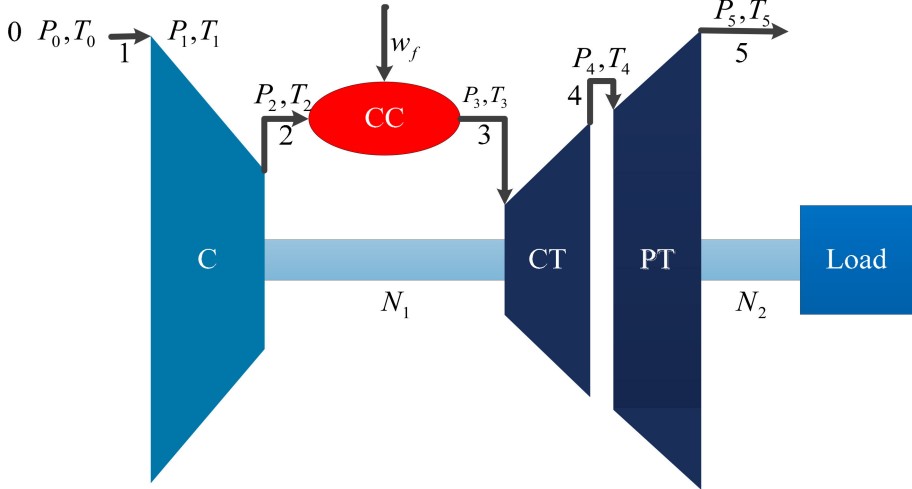

**Figure 3.** The layout of a two-shaft marine gas turbine. $T_0$ is the inlet temperature of the compressor and $P_0$ is the ambient pressure; $T_2$ and $P_2$ are the compressor outlet temperature and outlet pressure, respectively, and $N_1$ is the compressor speed; $w_f$ is the fuel mass flow which sprays into the combustion chamber to produce outlet temperature $T_3$ and outlet pressure $P_3$; $T_4$ and $P_4$ are the exhaust gas temperature and pressure of the compressor turbine, respectively; $T_5$ and $P_5$ are the outlet temperature and outlet pressure of the power turbine, respectively, and $N_2$ is the power turbine speed.

The nonlinear dynamic model of the two-shaft marine gas turbine was taken as a real engine, which was established according to the available literature [36], as shown in Equation (15); for detailed information refer to authors' previous work [35].

$$
\begin{aligned}
\dot{N}_1 &= \left(\frac{30}{\pi}\right)^2 \frac{1}{J_1 N_1}\left[m_{CT}\eta_m c_{pg}(T_3 - T_4) - m_C c_p(T_2 - T_1)\right] \\
\dot{N}_2 &= \left(\frac{30}{\pi}\right)^2 \frac{1}{J_2 N_2}\left[m_{PT} c_{pg}(T_4 - T_5) - \Phi N_2^3\right] \\
\dot{T}_3 &= \frac{R_g T_3}{P_3 V_1 c_{vg}}\left[k\left(c_{pg}T_2 m_C + LHV \eta_{CC} w_f - c_{pg}T_3 m_{CT}\right) - c_{pg}T_3\left(m_C + w_f - m_{CT}\right)\right] \\
\dot{P}_3 &= \frac{P_3}{T_3}\dot{T}_3 + \frac{R_g T_3 (m_C + w_f - m_{CT})}{V_1} \\
\dot{P}_4 &= \frac{(m_{CT} - m_{PT})R_g T_4}{V_2} \\
\dot{P}_5 &= \frac{(m_{PT} - \Gamma\sqrt{P_5})R_g T_5}{V_3}
\end{aligned}
\tag{15}
$$

where $J_1$ and $J_2$ are the inertia of the compressor shaft and power turbine shaft, respectively; $\Phi$ is the relationship coefficient between $N_1$ and the load; $LHV$ is the fuel low heating value; and $V_1$, $V_2$, and $V_3$ are the component volumes. $m_C$, $m_{CT}$, and $m_{PT}$ represent the mass inside the compressor, compressor turbine, and power turbine, respectively. $c_p$ and $c_{pg}$ represent the heat capacity of air and gas at constant pressure, respectively; $c_{vg}$ represents the volumetric heat capacity of gas; and $R_g$ denotes the gas constant. $\eta_m$ denotes the mechanical efficiency, and $\Gamma$ is the relationship coefficient between $m_{PT}$ and $P_5$.

Next, the performance of the FCM based FDI approach was tested in a simulation environment, as shown in Figure 4. According to Equation (15), there are six component performance parameters in a two-shaft marine gas turbine: compressor mass flow $m_C$, compressor efficiency $\eta_C$, compressor turbine mass flow $m_{CT}$, compressor turbine efficiency $\eta_{CT}$, power turbine mass flow $m_{PT}$, and power turbine efficiency $\eta_{PT}$. When the gas turbine is in a healthy condition, the compressor mass flow and efficiency are denoted $m_{C,H}$ and $\eta_{C,H}$, respectively; compressor turbine mass flow and efficiency are denoted $m_{CT,H}$ and $\eta_{CT,H}$, respectively; and power turbine mass flow and efficiency are denoted $m_{PT,H}$ and $\eta_{PT,H}$, respectively.

In this paper, gas path faults were simulated via changing component performance parameters. Typical single gas path faults used in this paper are shown in Table 2. When two or more single faults occurred simultaneously they were considered multiple faults. The sensor noise used in this paper was Gaussian noise, where the mean value was zero, and the relative standard deviation of each measurement variable is shown in Table 3.

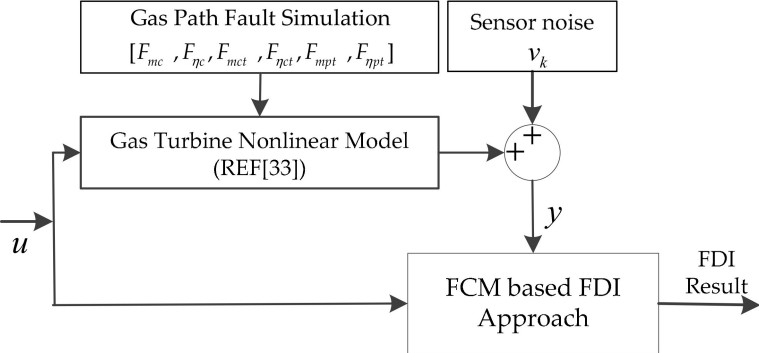

**Figure 4.** FCM based FDI approach simulation testing.

**Table 2.** Typical single gas path faults.

| Fault Description | Symbol | Fault Vector |
|---|---|---|
| Healthy condition | $H$ | $f_k = [0\,0\,0\,0\,0\,0]^T$ |
| Decrease in $m_C$ | $F_{mc}(b_1\%)$ | $f_k = [b_1\,0\,0\,0\,0\,0]^T$, $b_1 = (m_C/m_{C,H} - 1) \times 100\%$ |
| Decrease in $\eta_C$ | $F_{\eta c}(b_2\%)$ | $f_k = [0\,b_2\,0\,0\,0\,0]^T$, $b_2 = (\eta_C/\eta_{C,H} - 1) \times 100\%$ |
| Decrease in $m_{CT}$ | $F_{mct}(b_3\%)$ | $f_k = [0\,0\,b_3\,0\,0\,0]^T$, $b_3 = (m_{CT}/m_{CT,H} - 1) \times 100\%$ |
| Decrease in $\eta_{CT}$ | $F_{\eta ct}(b_4\%)$ | $f_k = [0\,0\,0\,b_4\,0\,0]^T$, $b_4 = (\eta_{CT}/\eta_{CT,H} - 1) \times 100\%$ |
| Decrease in $m_{PT}$ | $F_{mpt}(b_5\%)$ | $f_k = [0\,0\,0\,0\,b_5\,0]^T$, $b_5 = (m_{PT}/m_{PT,H} - 1) \times 100\%$ |
| Decrease in $\eta_{PT}$ | $F_{\eta pt}(b_6\%)$ | $f_k = [0\,0\,0\,0\,0\,b_6]^T$, $b_6 = (\eta_{PT}/\eta_{PT,H} - 1) \times 100\%$ |

**Table 3.** Sensor noise.

| Measurement Parameters | $N_1$ | $N_2$ | $T_2$ | $P_2$ | $T_4$ | $P_4$ | $T_5$ |
|---|---|---|---|---|---|---|---|
| Standard deviation (%) | 0.051 | 0.051 | 0.23 | 0.164 | 0.097 | 0.164 | 0.097 |

*4.1. FCM Based Model Set Testing*

In the two-shaft marine gas turbine model set, as shown in Equation (3), the state vector was $x = [N_1, N_2, T_3, P_3, P_4]^T$, the measurement vector $y = [N_1, N_2, T_2, P_2, T_4, P_4, T_5]^T$, and the control vector $u = w_f$. Sample time was 0.02 s.

4.1.1. Mode Set Accuracy Testing

When $f_k = 0$, the FCM based model set is a set of healthy linearized models. In this paper, the fitting approach based on perturbation [37] was used to establish the gas turbine linearized model, as shown in Equation (4), and gap metric analysis was used to select the operating point.

The range of the control vector $w_f$ was [0.3, 1]. With an equal interval of 0.01 for setting the operating points, the operating range was divided into 70 subranges, and a total of 71 gas turbine linearization models were established. According to Equation (14), the matrix of the gap metrics between the gas turbine linearization models was calculated and is shown in Figure 5.

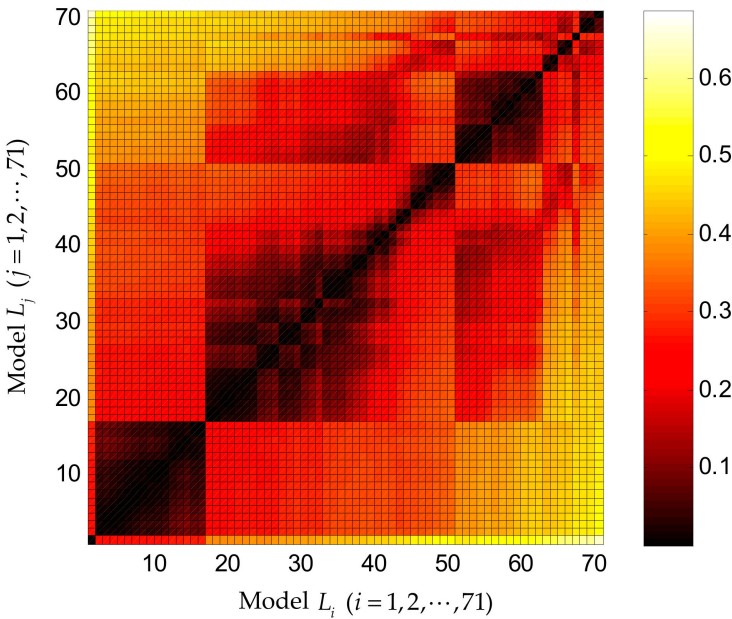

**Figure 5.** The matrix of the gap metric between the gas turbine linearized models.

The gap metric matrix is a symmetric matrix, and the diagonal elements are the gap metrics of each model with itself, hence the value is zero. The gas turbine operating range was segmented by comparing the pre-set threshold $\delta_{TH} = 0.25$ starting at the design operating point $w_f = 1$ (namely $L_1$), and six operating points were selected, as shown in Table 4, to yield the FCM based model set.

**Table 4.** The result of operating points selection based on the gap metric analysis.

| Models | 1# | 2# | 3# | 4# | 5# | 6# |
|--------|----|----|----|----|----|----|
| $i$ | 1 | 2 | 17 | 42 | 63 | 71 |
| $w_f$ | 1 | 0.99 | 0.84 | 0.59 | 0.38 | 0.3 |

The dynamic response of each measurement parameter was compared between the gas turbine nonlinear model [35] and FCM based model set. Figure 6 presents the corresponding differences between two models at a given fuel flow schedule. In this case, the fuel flow $w_f$ decreases gradually from 1.0 to 0.3 in the period from 0 to 40 s. It can be seen that errors at each selected operating point appear to be the largest for each subrange, which agrees with the operating point selection using the gap metric method. However, the absolute value of the maximum error for each measured parameter is less than 4%; that is, it is in an acceptable error range. At the expense of certain model simulation accuracy, the model set covering the wide operating condition only uses 8.5% of the operating points of gas turbine linearization models, which greatly reduces the storage space and calculation requirements of the model set.

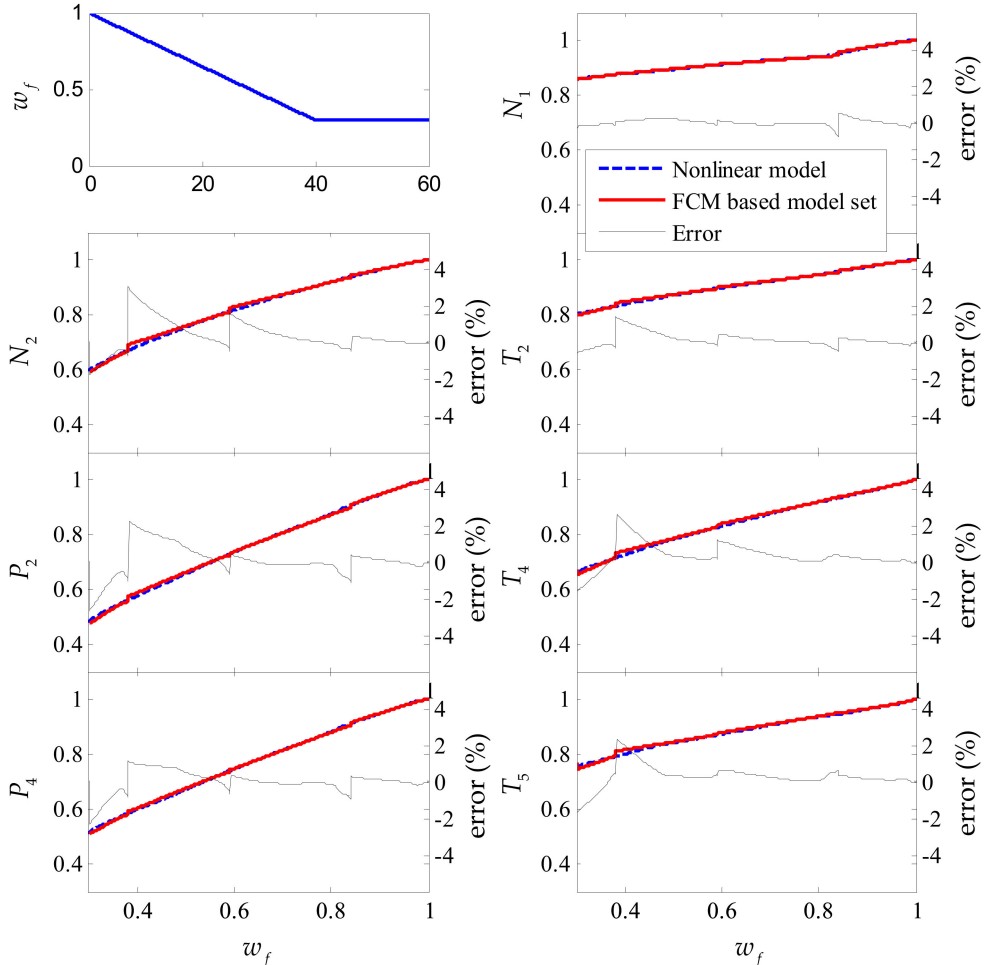

**Figure 6.** Comparison of the response of each measurement parameter between the nonlinear model and the FCM based model set at a given fuel flow schedule, and the corresponding error.

### 4.1.2. Hypothetical Fault Simulation

To verify whether the FCM based model set can be used to establish the hypothetical models, a comparative study was conducted with a nonlinear model in Equation (15), the traditional gas path fault model shown in Equation (2), and the hypothetical fault model auto-generated from the FCM based mode set, as shown in Equation (4). In this section, the traditional gas path model $F_{mc}$ at the operating point $w_f = 1.0$ is selected. Figure 7 shows the comparison dynamic tracking response in artificial simulation scenarios. The gas turbine was abruptly changed from operating point $w_f = 1.0$ to operating point $w_f = 0.99$ at $t = 15$ s; a $F_{mc}$ decrease of 1% occurs at $t = 15$ s and recovery to the healthy condition at $t = 25$ s; after 10 s, a $F_{mct}$ decrease of 1% occurs until the end.

It can be seen from Figure 7 that the hypothetical fault model auto-generated from the FCM based mode set has the same fault dynamic tracking response with a nonlinear model under a steady state condition and transient condition. The traditional gas fault model can only simulate the specified fault (here is $F_{mc}$). The hypothetical fault model auto-generated from FCM based mode set has adaptive fault tracking capability through adjusting $f_k$, which makes computation simpler than the traditional gas path fault model.

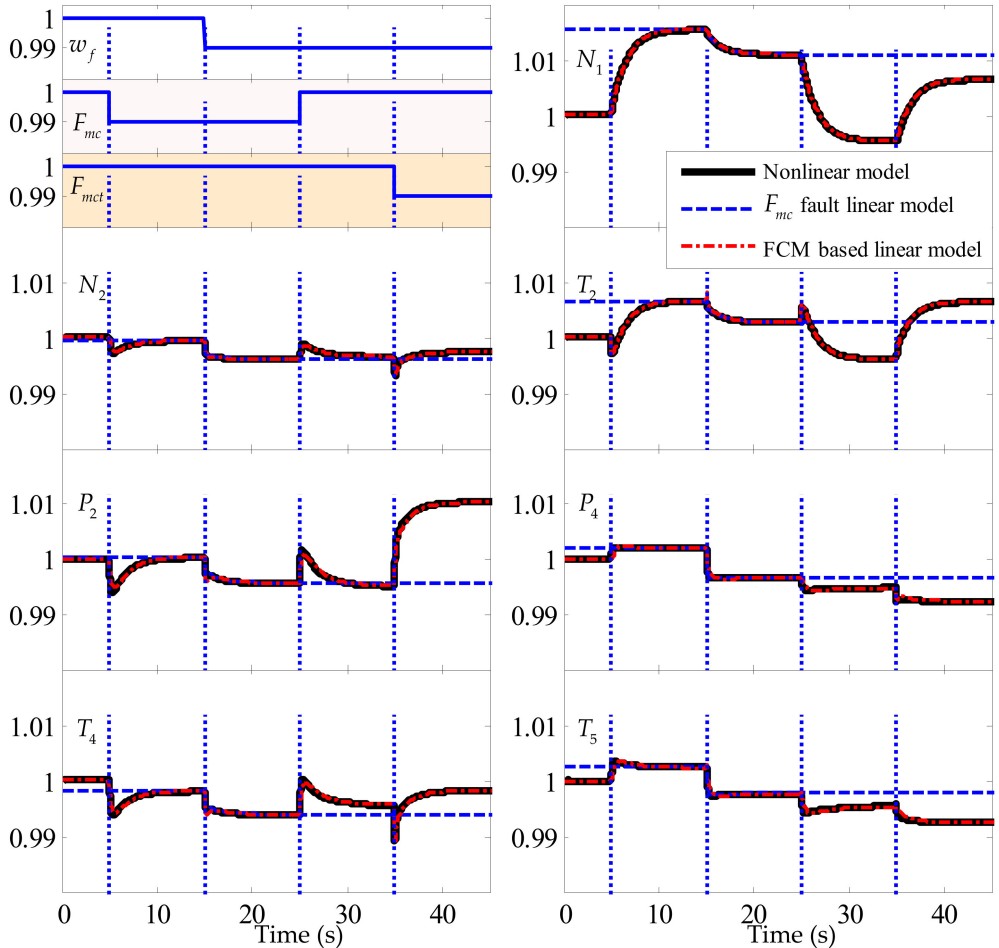

**Figure 7.** Performance comparison between the traditional $F_{mc}$ fault linear model at design operating points and the FCM based model set.

## 4.2. Single Fault Scenarios

### 4.2.1. FDI Results of a Single Fault under Different Operating Conditions

In this section, the gas turbine was worked under the steady condition at the operating point $w_f = 1.0$ until $t = 360$ s, and then the transient condition while the operating point dropped gradually to $w_f = 0.3$ at $t = 720$ s. Six gas path faults occurred separately in sequence under the steady condition and transient condition with an amplitude of $-1\%$, and the diagnosis results are shown in Figure 8.

It can be seen that the FCM based MM-FDI approach can accurately detect and isolate single gas path faults under both the steady and transient conditions at expected detection times and isolation periods, respectively.

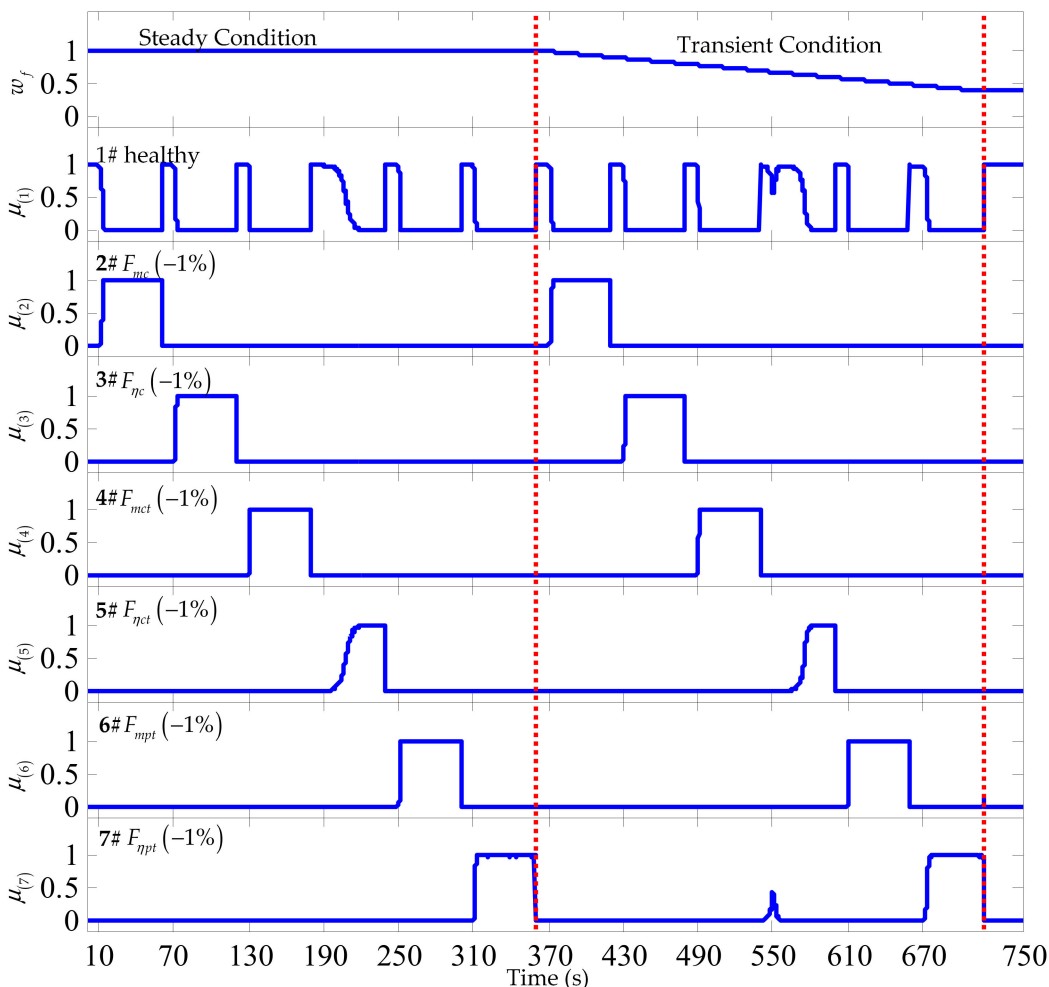

**Figure 8.** FDI result of a single gas path fault under the steady state condition and transient condition.

4.2.2. Performance under Different Single Fault Amplitudes

In this section, 100 simulations of single gas path faults were performed each time at the operating point, and repeated 10 times for a total of 1000 simulations based on the gas turbine nonlinear model. For each simulation, the type of fault was randomly generated, and the range of the fault amplitude $b_i$ was [−2.7%, −0.7%]. Keeping the hypothetical model set in an initial model sequence (fault amplitude $b_i$ is −1%), the correct detections (CD), the incorrect detections (ID), and the missed detections (MD) of the FCM based MM-FDI approach under a single fault were obtained from this Monte Carlo test, and are listed in Table 5.

**Table 5.** The fault detection and isolation (FDI) results of the 1000 stochastic simulations.

| | Confusion Matrix of the FDI Result | | | | | | Final Result | | |
|---|---|---|---|---|---|---|---|---|---|
| | $H$ | $F_{mc}$ | $F_{\eta c}$ | $F_{mct}$ | $F_{\eta ct}$ | $F_{mpt}$ | $F_{\eta pt}$ | CD | ID | MD |
| $H$ | 135 | 0 | 0 | 0 | 0 | 0 | 0 | | | |
| $F_{mc}$ | 0 | 123 | 0 | 0 | 0 | 0 | 0 | | | |
| $F_{\eta c}$ | 0 | 0 | 152 | 0 | 0 | 0 | 0 | | | |
| $F_{mct}$ | 0 | 0 | 0 | 121 | 0 | 0 | 0 | 95.5% | 0.45% | 0 |
| $F_{\eta ct}$ | 0 | 0 | 0 | 0 | 153 | 0 | 0 | | | |
| $F_{mpt}$ | 0 | 0 | 0 | 0 | 0 | 171 | 0 | | | |
| $F_{\eta pt}$ | 0 | 45 | 0 | 0 | 0 | 0 | 145 | | | |

It can be seen that the proposed approach allows 95.5% faults to be detected and isolated under the initial hypothetical mode set, which demonstrates that the proposed approach produces high diagnostic accuracy under different single fault scenarios.

All the incorrect detections are on $F_{\eta pt}$, which incorrectly detected as $F_{mc}$, as shown in Table 5. For an in-depth understanding, all the fault amplitudes of $F_{\eta pt}$ faults that were simulated in the case study and the corresponding FDI results are presented in Figure 9. It can be seen that all incorrect detection occurs when the fault amplitude is too large. Therefore, the cause of the incorrect detection is that the amplitude of the initial hypothetical $F_{\eta pt}$ model is small, and it does not match the actual fault condition. Fortunately, the large amplitude fault can be accurately detected and isolated when the hypothetical model set is updated, as advocated in this method.

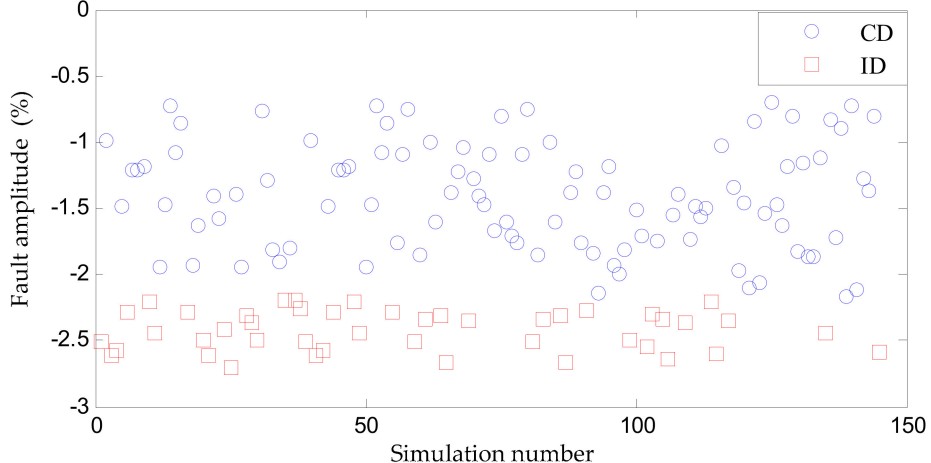

**Figure 9.** The fault amplitude of the $F_{\eta pt}$ fault and the corresponding FDI results of 145 simulations.

### 4.2.3. Performance under Different Numbers of Available Sensors

In practice, sensors often fail to provide effective measurement information, which results in the number of available sensors decreasing. Therefore, the influence of the number of available sensors on the performance of the proposed approach was studied at the operating point $w_f = 1.0$. Three cases were considered, seven sensors ($N_1$, $N_2$, $T_2$, $P_2$, $T_4$, $P_4$, and $T_5$), four sensors ($N_1$, $N_2$, $P_2$, and $T_4$), and two sensors ($N_1$ and $N_2$), and each fault had a 1% decrease in its output. The detection time and isolation time of each fault are shown in Table 6.

As shown in Table 6, the proposed approach can detect and isolate the fault with four sensors. However, several faults were not able to be differentiated, which are denoted as either MD or ID in the scenario of two available sensors. In addition, the fault detection time and the isolation time increases as the number of sensors decreases, which is mainly due to the reduction in the available information to distinguish different faults that the sensors can provide.

**Table 6.** The detection time and the isolation time for different numbers of measurements available.

| Fault Type | Seven Sensors | | Four Sensors | | Two Sensors | |
|---|---|---|---|---|---|---|
| | $t_d$ (s) | $t_i$ (s) | $t_d$ (s) | $t_i$ (s) | $t_d$ (s) | $t_i$ (s) |
| $F_{mc}$ | 1.46 | 5.02 | 4.8 | 17.65 | 223 | 1009 |
| $F_{\eta c}$ | 0.9 | 3.68 | 4.78 | 51.4 | MD/ID | MD/ID |
| $F_{mct}$ | 0.28 | 0.9 | 0.7 | 5.02 | 912 | 3739 |
| $F_{\eta ct}$ | 4.32 | 17.9 | 2.92 | 62.8 | MD/ID | MD/ID |
| $F_{mpt}$ | 0.34 | 1.2 | 5.12 | 25.7 | MD/ID | MD/ID |
| $F_{\eta pt}$ | 0.54 | 1.8 | 2.08 | 8.5 | 22.5 | 116.2 |

### 4.2.4. Performance under Measurement Outliers

It is inevitable to have outliers in measurements due to various interferences, such as failed communication, hardware malfunction, or environment change, which can result in a false alarm. Therefore, some gas path diagnosis methods have been proposed for pre-processing the measurement signals to remove these outliers and noise before FDI. In this case, the outliers were artificially added to the measurement signals and the outliers were selected at the 2% level of the nominal value of these measurements. Suppose a $F_{mc}$ decrease of 1% occurs at $t = 20$ s, the variation of each measurement parameter and the probability of each hypothetical model are shown in Figure 10. It can be seen that the proposed approach can accurately detect and isolate faults when there are outliers, and the outliers only slightly disturb the conditional probability of the model. Therefore, the FCM-based FDI approach proposed in this paper is robust in the measurement of outliers.

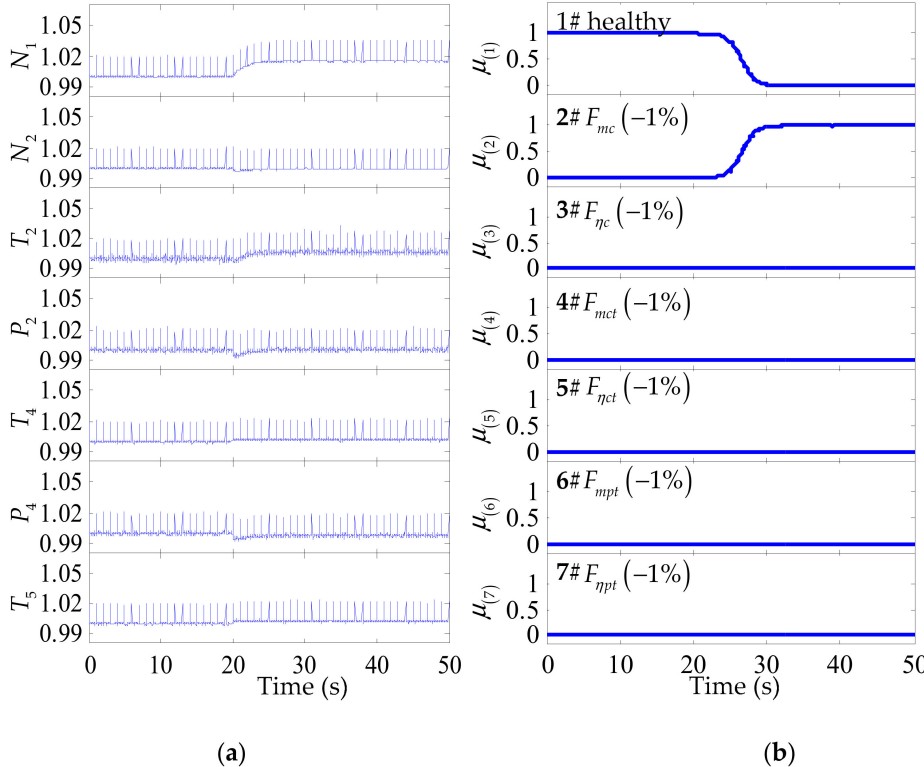

(**a**)  (**b**)

**Figure 10.** Performance of the FCM based MM-FDI approach under outliers. (**a**) The measurement parameters in the scenarios of $F_{mc}$ decreasing by 1% at $t = 20$ s; (**b**) The conditional probability $\mu_i$ of each hypothetical model in the scenarios of $F_{mc}$ decreasing by 1% at $t = 20$ s along with outliers.

### 4.3. Multiple Fault Scenarios

#### 4.3.1. FDI Results of Multiple Faults in the Gas Path

Figure 11 shows the FDI results for the scenarios of a $-2\%$ change in $F_{\eta pt}$ at $t = 25$ s and a 1% change in $F_{\eta c}$ at $t = 35$ s under operating point $w_f = 1.0$. In this case, the hypothetical model set update approach was used for detecting and isolating these faults. It can be seen that when a $F_{\eta pt}$ decreasing by 2% fault occurs, the conditional probability of the 7# hypothetical model in Figure 11a is the largest and exceeds $\mu_{TH}$, which means this hypothetical model is closest to the actual fault condition. Then, the hypothetical model set is updated and the conditional probability is reinitialized correspondingly. With the updated hypothetical model set, the 7# hypothetical model, as shown in Figure 11b, is closest to the actual fault condition, so the conditional probability of the model is largest and over $\mu_{TH}$. Then, the hypothetical model set will be updated again and the conditional probability will be reinitialized again. Finally, as shown in Figure 11c, the $F_{\eta pt}$ and the $F_{\eta c}$ faults are accurately detected and isolated.

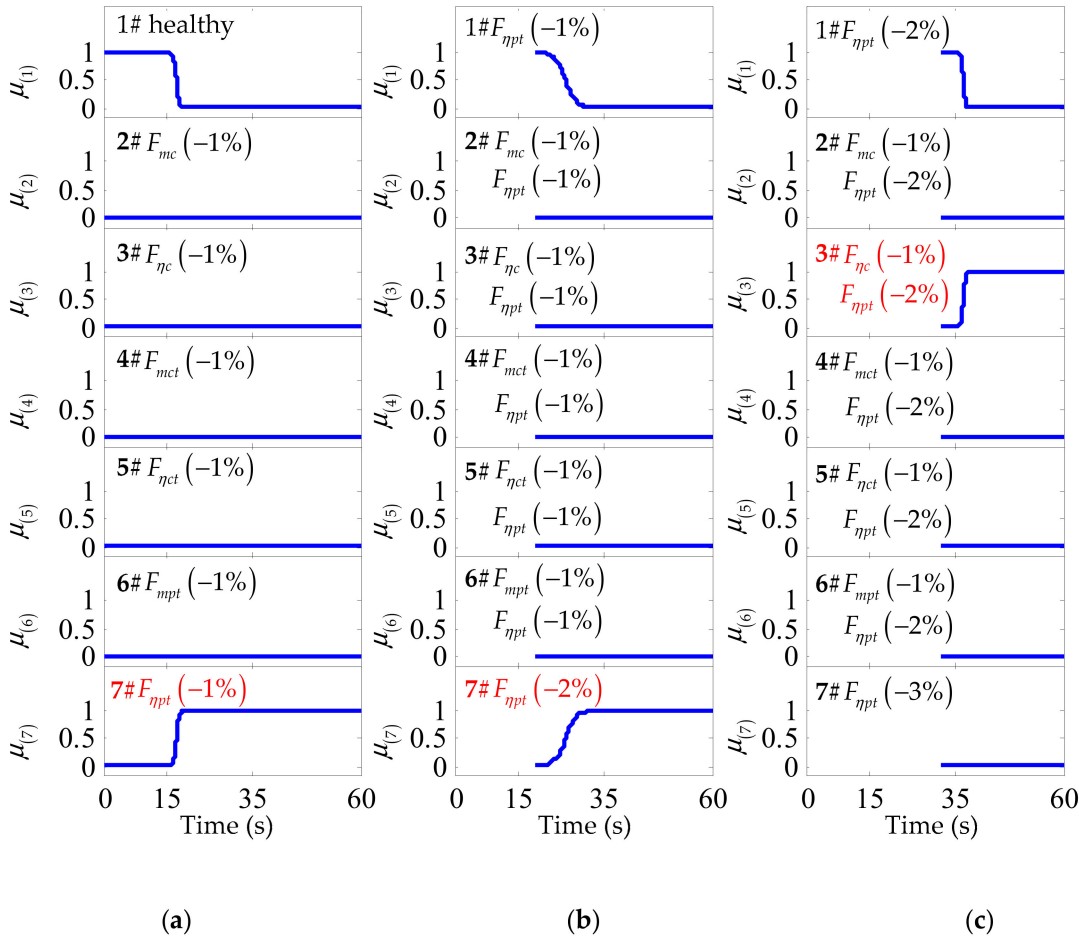

**Figure 11.** The multiple fault detection and isolation results for the scenarios of a 2% decrease in $F_{\eta pt}$ at $t$ = 15 s and 1% decrease in $F_{\eta c}$ at $t$ = 35 s. (**a**) The conditional probability of the initial model set; (**b**) The conditional probability of the updated model set 1; (**c**) The conditional probability of the updated model set 2.

### 4.3.2. Performance under Multiple Faults

A simulation was also conducted for the scenario of multiple faults at the operating point $w_f = 1.0$, and the detection and isolation results are shown in Figure 12 and Table 7. In this scenario, the type of fault was random and the range of the fault amplitude $b_i$ was [−2.7%, −0.7%]. In addition, the fault amplitude of the gas path fault caused by fouling, corrosion, and erosion may be gradual, so both abrupt and gradual changes in fault amplitude were considered. In the case of an abrupt fault, the first fault occurs at $t$ = 20 s and the second fault occurs at $t$ = 70 s. In the case of a gradual fault, the fault amplitude of both faults changes from $t$ = 0 s, with the first fault changing to a given amplitude at $t$ = 20 s, and the second fault changing to a given amplitude at $t$ = 70 s.

It can be seen that, regardless of whether the fault amplitude changes abruptly or gradually, the first fault was 100% accurately detected and isolated using the proposed approach. In the case of an abrupt fault, the second fault shows five faults were incorrectly identified as three or more faults, and one fault was missed due to its very small amplitude. In the case of a gradual fault, the second fault shows that five faults were incorrectly identified to be three faults. Therefore, the proposed approach can effectively improve the FDI performance of large amplitude single faults, and has high FDI performance with 94% and 95% accuracy in the multiple abrupt faults and gradual faults scenarios, respectively.

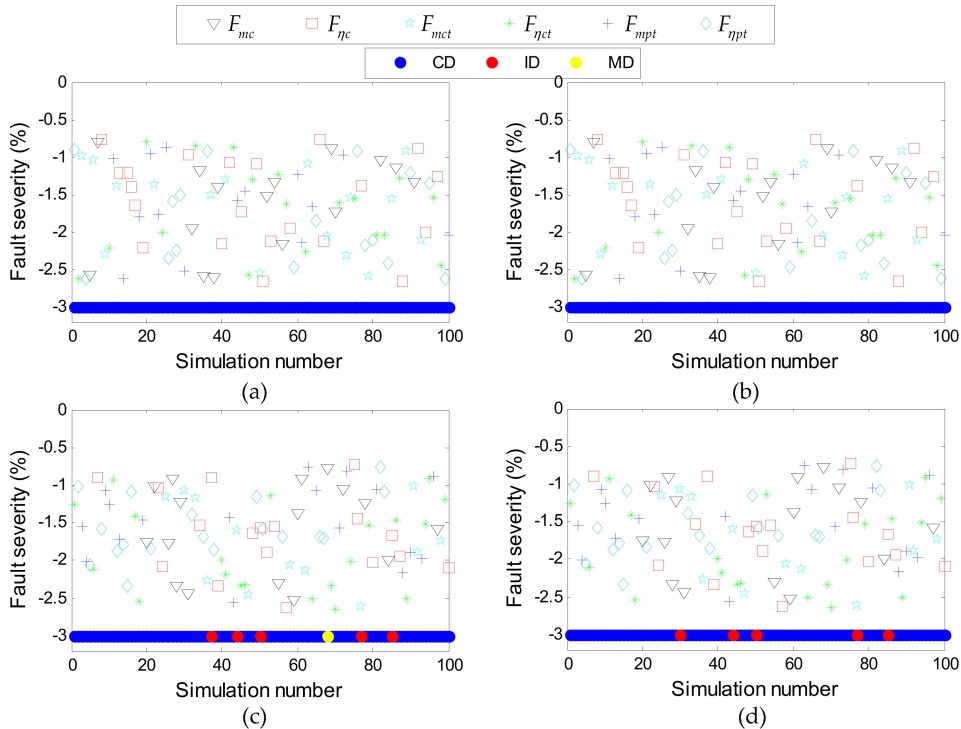

**Figure 12.** The type and amplitude of multiple faults and the corresponding FDI results of 100 stochastic simulations. (**a**) The type and amplitude of the first abrupt fault and corresponding FDI results; (**b**) The type and amplitude of the first gradual fault and corresponding FDI results; (**c**) The type and amplitude of the second abrupt fault and corresponding FDI results; (**d**) The type and amplitude of the second gradual fault and corresponding FDI results.

**Table 7.** The FDI results of 100 stochastic simulations in the scenarios of multiple faults.

| Fault Change | Faults | Simulation Times | FDI Result | | |
|---|---|---|---|---|---|
| | | | CD | ID | MD |
| Abrupt | First fault | 100 | 100 | 0 | 0 |
| | Second fault | 100 | 94 | 5 | 1 |
| | Total | 100 | 94 | 5 | 1 |
| Gradual | First fault | 100 | 100 | 0 | 0 |
| | Second fault | 100 | 95 | 5 | 0 |
| | Total | 100 | 95 | 5 | 0 |

### 4.3.3. FDI Results of Multiple Faults in both the Actuator and Gas Path

In Equation (3), gas path faults are converted into a type of fault in the model set that is similar to the actuator fault. Due to the good scalability of the FCM based FDI approach, it is easy to apply the proposed approach to gas path and actuator multiple faults.

In this section, the actuator fault corresponds to the fuel flow ($F_{wf}$). The fuel actuator gets stuck at some value [38] $a_0 = \left( w_f / w_{f,H} - 1 \right) \times 100\%$, which is the fuel actuator fault $F_{wf}(a_0\%)$; $u_k = m_k + a_0$, where $m_k$ is controller fuel mass output. Considering the healthy condition and six gas path faults, as shown in Table 2, there will be eight hypothetical models in the model set. Although the number of hypothetical models has increased, the process of the FDI of multiple faults in both the gas path and actuator is consistent with the process of the FDI of gas path faults.

Figure 13 shows the FDI results in the scenarios of the actuator fault, where the $F_{wf}$ decreases by 5% fault occurs at $t = 15$ s, the $F_{mc}$ decreases by 1% fault occurs at $t = 35$ s, and the $F_{\eta pt}$ decreases by 1% fault occurs at $t = 45$ s. In these scenarios, the gas turbine engine works at the operating point

$w_f = 1.0$. Assume that the initial fault amplitude of $F_{wf}$ is $-5\%$ and the gas path fault amplitude is $-1\%$. The FDI results show that both the gas path fault and actuator fault are accurately detected and isolated, which indicates that the occurrence of an actuator fault will not affect the detection and isolation of a gas path fault.

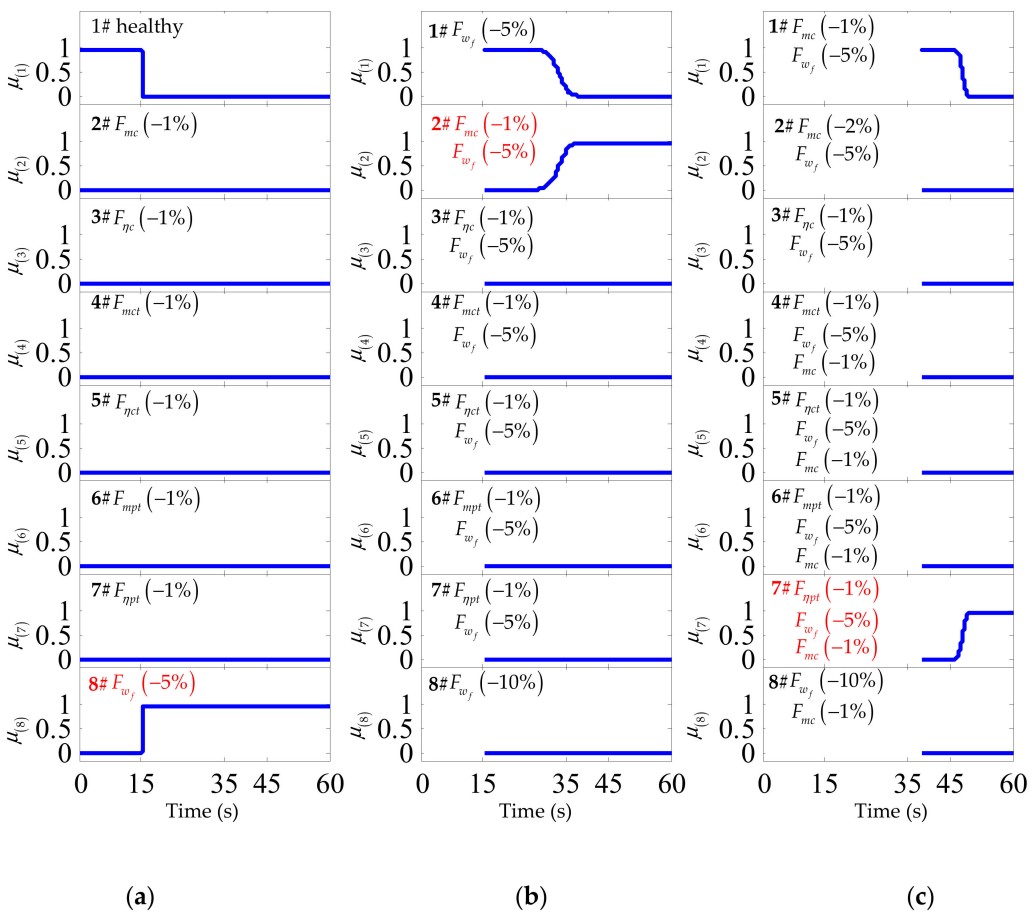

(**a**)                    (**b**)                    (**c**)

**Figure 13.** The multiple faults detection and isolation results in the scenarios of $F_{wf}$ decreases by 5% fault occurs at $t = 15$ s, $F_{mc}$ decreases by 2% fault occurs at $t = 35$ s, and $F_{\eta pt}$ decreases by 1% fault occurs at $t = 45$ s. (**a**) The conditional probability of the initial model set; (**b**) The conditional probability of the updated model set 1; (**c**) The conditional probability of the updated model set 2.

## 5. Conclusions

In this paper, an FCM based MM-FDI approach was developed to detect and isolate the gas path faults of gas turbines over a wide operating condition. It shows that the FCM can be realized via an additive term of the healthy model set, allowing hypothetical models for various gas path faults to be easily established and updated online. A gap metric analysis for operating points selection converts the nonlinearity of the gas turbine into multiple linearized models over a wide range of operating conditions with specified accuracy and computational efficiency. Several simulation case studies on a two-shaft marine gas turbine were conducted, and the performance of the proposed approach for the FDI of a single fault and multiple faults, and the influence of measurement outliers, was analyzed. The simulation results show that the proposed FCM based MM-FDI approach realizes the adaptive generation of the fault model set by taking the gas path fault as an additive term and extending the gas path fault matrix into the control vector, which makes MM-FDI approaches achievable and efficient for online gas path fault detection and isolation application. The gap metric analysis for selecting operating points of the healthy model set greatly reduces the storage space and calculation requirement of the model set.

**Author Contributions:** This research was completed through a collaboration of all authors. Q.Y. and Y.C. proposed the FCM based MM-FDI approach and wrote the original manuscript; S.L. was the team leader of this work and responsible for coordination; F.G., A.S., and Y.C. designed the case studies and analyzed the results.

**Funding:** This research was funded by the Fundamental Research Funds for the Central Universities [HEUCFP201722].

**Acknowledgments:** Gratitude is extended to Harbin Engineering University for supporting Y.C. to carry out collaborative research in the Centre for Efficiency and Performance Engineering at the University of Huddersfield.

**Conflicts of Interest:** The authors declare no conflict of interest.

## Nomenclature

*Symbols*

| | |
|---|---|
| $A_i$ | state matrix of the healthy condition at the operating point $i$ |
| $A_i'$ | state matrix of a gas path fault condition at the operating point $i$ |
| $B_i$ | control matrix of the healthy condition at the operating point $i$ |
| $B_i'$ | control matrix of a gas path fault condition at the operating point $i$ |
| $C_i$ | output matrix of the healthy condition at the operating point $i$ |
| $C_i'$ | output matrix of a gas path fault condition at the operating point $i$ |
| $D$ | gas path fault contribution matrix for the output vector |
| $E$ | gas path fault contribution matrix for the state vector |
| $G$ | the gap metric matrix of all linearized models |
| $Hm_i$ | the $i$-th hypothetical model |
| $J$ | inertia of shaft |
| $K$ | Kalman gain |
| $L_i$ | the $i$-th linear model |
| $M_i, N_i$ | the normalized right coprime factorizations of linear model $L_i$ |
| $N$ | rotational speed |
| $P$ | pressure |
| $Q$ | process noise covariance |
| $R$ | measurement noise covariance for FDI |
| $R_g$ | gas constant |
| $S$ | innovation covariance |
| $T$ | temperature |
| $V$ | component volume |
| $W$ | the number of hypothetical models |
| $b$ | fault amplitude |
| $c_p$ | air constant pressure specific heat |
| $c_{pg}$ | gas constant pressure specific heat |
| $c_{vg}$ | gas constant volume specific heat |
| $f$ | gas path fault vector |
| $k$ | adiabatic index |
| $l$ | the dimension of the measurement vector |
| $m$ | gas/air mass flow of component |
| $n$ | the dimension of the state vector |
| $q$ | number of gas path faults |
| $s$ | fault amplitude changes during model set updating |
| $t$ | the dimension of the control vector |
| $u$ | control vector |
| $v_k$ | measurement noise |
| $w_k$ | process noise |
| $w_f$ | fuel mass flow |
| $x$ | state vector |
| $y$ | measurements vector |
| $z$ | fault location vector |

*Greek*

| | |
|---|---|
| $\Gamma$ | coefficient between mass flow and pressure |
| $\Phi$ | coefficient between $N_2$ and load |
| $\gamma$ | filter innovation |
| $\delta\left(L_i, L_j\right)$ | gap metric between linear models $L_i$ and $L_j$ |
| $\delta_{TH}$ | the preset threshold of the gap metric |
| $\eta$ | isentropic efficiency |
| $\mu_{TH}$ | the preset threshold of the conditional probability |
| $\mu_i$ | the conditional probability of the $i$-th hypothetical model |
| $\varphi$ | the interval between any two adjacent operating points |

*Subscript*

| | |
|---|---|
| $C$ | compressor |
| $CC$ | combustion chamber |
| $CT$ | compressor turbine |
| $H$ | health condition |
| $PT$ | power turbine |
| $k$ | discrete time $k$ |

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
