# Peer review of "A Gas Path Fault Contribution Matrix for Marine Gas Turbine Diagnosis Based on a Multiple Model Fault Detection and Isolation Approach"

_energies, doi:10.3390/en11123316_

Round 1

Reviewer 1 Report

This paper is interesting and contributes to the characterization and  predition of possible gas path fault for Marine Gas Turbine. The model is able to reproduce with confinable results the effect typical gas path fault as decrease of: quality of the fuel, fuel flow, efficiency of compressor and turbine, etc., using the normal measurements for Gas Turbines systems, rotation speeds, pressures and temperatures of the flow.

In my opinion the model is good but it could be improved, take into account the evolution of the combustion in the combustion chamber, I am sure that including the parameter of efficiency combustion chamber the results will be improve.

In other hand, I recommend for future works, the use of a more detailed model for the Gas Turbine system, this model could be included a combustion modellitation, effect over the triangle of velocities, etc. An important contribution for detect fault using a thermodinamical model is describe in this recommend paper Marine diesel engine failure simulator based on thermodynamic model (https://doi.org/10.1016/j.applthermaleng.2018.08.096)

Firstly, the manuscript is quite good presented with the figures and tables included in correct place, it is very easy to follow the dissertation. I have not found writing or grammar mistakes.

The mathematical equations include numeration easy to follow in a completed reading, but when it is reading in separate sessions, it is difficult to found the meaning of all parameter. An independent and separate lists of nomenclature, abbreviations and symbols is recommended in order to follow correctly the acronyms uses in the text and the meaning of the variables for each equations.

I think that the article is valid to be printed in the Energies Journal but I recommend to improve a little the paper with a table with the nomenclature.

Reviewer 2 Report

TITLE: A Gas Path Fault Contribution Matrix for Marine Gas Turbine Diagnosis Based on a Multiple Model Fault Detection and Isolation Approach

AUTHORS: Qingcai Yang, Shuying Li, Yunpeng Cao, Fengshou Gu and Ann Smith

(Manuscript energies-381548)

Reviewer comments:

In the paper, in order to improve the gas turbines components fault detection and isolation (FDI), by the multiple model (MM) approaches, the author presents a fault contribution matrix (FCM) technique, well described in the article. The positive results in the single and multiple faults detection of a two-shafts marine gas turbine, presented in the paper, proves the validity of the proposed solution.

The article, substantially well organized and written, requires minor corrections and improvements:

-         the bibliographic analysis can be improved with the paper:

Campora U., Cravero C., Zaccone R. Marine Gas Turbine Monitoring and Diagnostics by Simulation and Pattern Recognition. International Journal of Naval Architecture and Ocean Engineering, Publisher: Society of Naval Architects of Korea, Volume 10, Issue 5, September 2018, pp 617-628. DOI: 10.1016/j.ijnaoe.2017.09.012.

Where Mahalanobis distance and artificial neural networks techniques are combined to marine gas turbine faults diagnostics;

-         at the beginning of line 281 replace 'the' with 'The';

-         in line 312 replace ‘single fault Scenarius’ with ‘Single fault scenarius’;

-         in line 341 replace ‘Figure 12’ with ‘Figure 9’;

-       in line 372 replace ‘multiple faults Scenarius’ with ‘Multiple faults scenarius’;

-         in line 374 replace ‘Figure 9’ with ‘Figure 11’.

Is reviewer opinion that, with the proposed improvements, the paper is acceptable for the publication in the Energies magazine.

Reviewer 3 Report

The current paper presents an FCM based MM-FDI approach to detect and isolate the gas
path faults in gas turbines over a wide operating condition. Several simulation case studies on a two-shaft marine gas turbine were conducted, and the performance of the proposed approach for
the FDI of single fault and multiple faults and the influence of measurement outliers were analyzed. The paper's subject has a high engineering value. The methodology is well presented and discussed and the conclusions are supported by the results. There are adequate references of past related work. Overall, it is a good paper that merits publication in the Journal.
